# Determining Optimal Cut-Off Value of Pancreatic-Cancer-Induced Total Cholesterol and Obesity-Related Factors for Developing Exercise Intervention: Big Data Analysis of National Health Insurance Sharing Service Data

**DOI:** 10.3390/cancers15225444

**Published:** 2023-11-16

**Authors:** Hyunseok Jee, Kim Sang Won

**Affiliations:** 1School of Kinesiology, Yeungnam University, 280, Daehak-ro, Gyeongsan 38541, Republic of Korea; 2Medical Research Center, College of Medicine, Yeungnam University, Daegu 42415, Republic of Korea; kimsw3767@ynu.ac.kr

**Keywords:** pancreatic cancer, big data analysis, National Health Insurance Sharing Service Database, prevention program development, obesity

## Abstract

**Simple Summary:**

This study aimed to investigate the impact of several variables affecting the development of pancreatic cancer through the analysis of a sample cohort of 1,108,369 individuals from the National Health Insurance Sharing Service (NHISS DB). Furthermore, we aimed to develop individualized evidence-based prognostic and preventive programs (e.g., tailored exercise-based programs) through the derivation of cut-off value results of significant parameters to improve pancreatic cancer. There were differences between the pancreatic cancer versus non-pancreatic cancer groups in terms of gender; for example, body mass index, systolic/diastolic blood pressure, fasting blood glucose, and total cholesterol concentrations were lower in women with pancreatic cancer than in women without pancreatic cancer (*p* < 0.01). Fasting blood glucose and total cholesterol concentrations were significantly different between men with versus without pancreatic cancer (*p* < 0.05). Logistic regression analysis identified more than 20 parameters, including the impact of exercise. Afterward, the ROC curve analysis revealed significant cut-off values, which differed between men and women (i.e., the ROC curve analysis showed that total cholesterol concentration was the only significant factor associated with pancreatic cancer in men). Our analysis of large-scale data from the NHISS DB suggests that identifying significant parameters helps in developing individualized evidence-based prognostic and preventive programs (e.g., tailored exercise-based programs) for ameliorating pancreatic cancer.

**Abstract:**

This study aimed to examine the effects of multiple parameters on the incidence of pancreatic cancer. We analyzed data from 1,108,369 individuals in the National Health Insurance Sharing Service Database (NHISS DB; birth to death; 2002 to 2015) and identified 2912 patients with pancreatic cancer. Body mass index, systolic/diastolic blood pressure, and fasting blood glucose and total cholesterol concentrations were lower in women with than without pancreatic cancer (*p* < 0.01). Fasting blood glucose and total cholesterol concentrations were significantly different between men with and without pancreatic cancer (*p* < 0.05). In the logistic regression analysis, the total cholesterol concentration (odds ratio (OR), 1.007; 95% confidence interval (CI), 1.005–1.010) was significantly higher in men than women with pancreatic cancer (*p* < 0.05). Pancreatic cancer rates were highest in men who smoked for 5–9 years or more (OR, 5.332) and in women who smoked for 10–19 years (OR, 18.330). Daily intensive exercise reduced the risk of pancreatic cancer by 56% in men (95% CI, 0.230–0.896). Receiver operating characteristic curve analysis revealed a total cholesterol concentration cut-off point of 188.50 mg/dL (*p* < 0.05) in men with pancreatic cancer, with a sensitivity and specificity of 53.5% and 54.6%, respectively. For women, the cut-off values for weight and gamma glutamyl transpeptidase concentration were 58.5 kg and 20.50 U/L, respectively. The sex-specific differences in patients with pancreatic cancer identified herein will aid in the development of individualized evidence-based prognostic and preventive programs for the treatment of pancreatic cancer.

## 1. Introduction

Pancreatic cancer has one of the highest mortality and incidence rates, which can be attributed to increased levels of carcinoembryonic antigen and carbohydrate antigen 19–9 [1]. The incidence of pancreatic cancer is highly associated with age > 50 years, dietary habits, and smoking [2]. Every year, 217,000 people are diagnosed with pancreatic cancer globally, and 213,000 patients die of pancreatic cancer. The 5-year survival rate for pancreatic cancer ranges from 2% to 9%, depending on the country, with regional and income differences; however, it does not exceed 10% [3].

Overall, cancer rates may differ depending on the country; however, these differences may not be significant because these rates vary according to age, sex, and cancer type [4,5]. For example, the incidence of colon cancer in men and women is significantly associated with other factors, such as fiber intake [6]. Previous studies have reported sex-specific differences in the incidence rates of various cancer types [7,8].

A previous study reported that those free of prior cancer at baseline with a body mass index (BMI) ≥ 30 kg/m^2^ have about 70% higher risk of pancreatic cancer compared to those with a BMI < 23 kg/m^2^ [9], indicating that the regulation of muscle mass or obesity is directly associated with carcinogenesis in the pancreas [10]. An epidemiological study also reported that physical activity considerably decreases pancreatic cancer incidence by 28% [11], indicating that exercise has anti-cancer effects [12]. However, previous studies have provided limited data on the effects of increased physical activity. Therefore, further studies should be conducted on appropriate exercise interventions (including the ideal intensity and duration of exercise) based on the type of cancer.

All Koreans are required to register their disease histories during regular checkups in the National Health Insurance Sharing Service Database (NHISS DB; https://www.nhis.or.kr) [13]. The NHISS DB supports longitudinal, cross-sectional, retrospective, prospective, and panel data-based studies [14].

In this study, we aimed to examine the effects of the parameters included in the NHISS DB on pancreatic cancer incidence based on multiple aspects. The demographic characteristics of patients with pancreatic cancer were evaluated based on data from 1,108,369 individuals in the NHISS DB. By analyzing sex-specific parameters associated with pancreatic cancer using Student’s *t*-tests, logistic regression, and receiver operating characteristic (ROC) curve analyses, useful information can be obtained for planning scientific, evidence-based interventions to improve the prognosis or preventive management of pancreatic cancer. The obtained information can also be used to develop exercise-based interventions.

## 2. Materials and Methods

### 2.1. Data Source and Subject Population

We used the NHISS Sample Research DB (https://nhiss.nhis.or.kr/bd/ab/bdaba000eng.do, accessed on 25 October 2022). The DB is sample cohort data for research purposes collected and accumulated from existing nationwide surveys and hospital prescription results. For this study, we used publicly available data, which were fully surveyed and finalized into a dataset by applying inclusion/exclusion criteria for the 1 million cohort data. The National Health Insurance Service DB is not customized data centered on specific diseases but a DB sampled from the population, which received health checkups regardless of disease, i.e., it is sample data of the 1 million cohort; therefore, it is sample data of the entire population. The sample survey DB contains 1,108,369 patient records, which is about 2% of the total population of South Korea. These 1,108,369 randomly selected patients are designed to be statistically representative of the entire population of South Korea (approximately 50 million people) and can be filtered by sex, age, member classification, medical insurance, region, and other strata. The number of people in the sample cohort data may vary slightly, as there is a 2% oversampling to reflect additional deaths and newborns each time. Patients who underwent regular health checkups from birth to death were randomly selected. All Koreans mandatorily registered in the NHISS DB were traced from 2002 to 2015. Pancreatic cancer patients were identified according to the Korean Standard Classification of Disease and Cause of Death (KSCDCD) codes (http://kssc.kostat.go.kr/ksscNew_web/index.jsp, accessed on 25 October 2022; C25, malignant neoplasm of the pancreas; C25.0, malignant neoplasm of the head of the pancreas; C25.1, malignant neoplasm of the body of the pancreas; C25.2, malignant neoplasm of the tail of the pancreas; C25.4, malignant neoplasm of the endocrine pancreas; C25.7, malignant neoplasm of other parts of the pancreas; C25.8, malignant neoplasm, overlapping lesion of the pancreas; C25.9, malignant neoplasm of the pancreas, unspecified; D01.70, carcinoma in situ of the pancreas; M8150, pancreatic endocrine tumor; and M8154, mixed pancreatic endocrine and exocrine tumor).

The Institutional Review Board of Yeungnam University (7002016-E-2022-015-01, approval date: 27 July 2022) approved this study as an exemption because all participants were anonymized, such that their ID numbers could be traced to secure their personal information.

### 2.2. Study Design

In total, 2912 patients with pancreatic cancer were identified from the NHISS DB using the aforementioned codes. Patients with diseases other than the common cold were excluded from the control group. All Korean patients and those with complaints were registered, and we aimed to determine the healthier group. Similarly, J00 (acute nasopharyngitis) was used as the code for the common cold. Sex-specific measurements of selected parameters were performed using Student’s *t*-tests. Factors significantly influencing pancreatic carcinogenesis were assessed using logistic regression analyses, and the optimal threshold points were determined (Figure 1).

### 2.3. Categorization of Variables

The following variables were examined: INDHIS1, individual disease history 1 (1, tuberculosis; 2, hepatitis; 3, hepatism; 4, high blood pressure; 5, cardiopathy; 6, cerebral apoplexy; 7, diabetes; 8, cancer; 9, others); INDHIS2, individual disease history 2 (1, tuberculosis; 2, hepatitis; 3, hepatism; 4, high blood pressure; 5, cardiopathy; 6, cerebral apoplexy; 7, diabetes; 8, cancer; 9, others); INDHIS3, individual disease history 3 (1, tuberculosis; 2, hepatitis; 3, hepatism; 4, high blood pressure; 5, cardiopathy; 6, cerebral apoplexy; 7, diabetes; 8, cancer; 9, others); SMK, smoking period in the past (1, within 6 years; 2, 6–9 years; 3, 10–19 years; 4, 20–29 years; 5, >30 years); EXERCI, exercise frequency (1, no exercise; 2, 1–2 times a week; 3, 3–4 times a week; 4, 5–6 times a week; 5, almost every day); HT, height (cm); WT, weight (kg); BMI, body mass index (kg/m^2^); SBP, systolic blood pressure (mm Hg); DBP, diastolic blood pressure (mm Hg); URIPRO, protein in urine (1, negative; 2, weak positive; 3, positive [+1]; 4, positive [+2]; 5, positive [+3]); HMG, hemoglobin (g/dL); FBS, fasting blood sugar concentration (mg/dL); TC, total cholesterol concentration (mg/dL); SGOT, serum glutamic oxaloacetic transaminase and aspartate aminotransferase (U/L); SGPT, serum glutamic pyruvic transaminase and alanine aminotransaminase concentration (U/L); GTP, gamma glutamyl transpeptidase concentration (U/L); AGE, age (years); CANCERHIS, individual disease history (1, tuberculosis; 2, hepatitis; 3, hepatism; 4, high blood pressure; 5, cardiopathy; 6, cerebral apoplexy; 7, diabetes; 8, cancer; and 9, others); FAMHIS, familial disease history (1, hepatism; 2, high blood pressure; 3, cerebral apoplexy; 4, cardiopathy; 5, diabetes; 6, cancer or others); VPA, number of days completing 20 min of intensive exercise in a week (1, 1 d; 2, 2 d; 3, 3 d; 4, 4 d; 5, 5 d; 6, 6 d; 7, 7 d); MPA, number of days completing 30 min of moderate exercise in a week (1, 1 d; 2, 2 d; 3, 3 d; 4, 4 d; 5, 5 d; 6, 6 d; 7, 7 d); LPA, number of days completing ≥ 30 min of walking in a week (1, 1 d; 2, 2 d; 3, 3 d; 4, 4 d; 5, 5 d; 6, 6 d; 7, 7 d); SICK1, primary sickness symptom; and SICK2, secondary sickness symptom.

### 2.4. Statistical Analysis

All data are presented as the mean ± standard deviation. A Student’s t-test was used to analyze the sex-specific differences between the cancer and non-cancer groups. Logistic regression was used to analyze the effects of sex on different pancreatic cancer incidence variables based on the odds ratios (ORs) and 95% confidence intervals (CIs). The effects of different exercise modes on pancreatic carcinogenesis were analyzed using logistic regression analysis. The ROC curve analysis was used to determine the optimal threshold for significant parameters identified in the logistic regression analysis. SAS version 9.4 (SAS Institute, Cary, NC, USA) and R (version 4.3.1, http://www.r-project.org, accessed 25 October 2022) were used for all statistical analyses. Statistical significance was set at *p* < 0.05 for all analyses.

## 3. Results

### 3.1. Demographic Characteristics

A total population of 1,108,369 individuals (556,583 men and 551,786 women), representative of the overall Korean population, was included in this study (Table 1). Individuals were registered based on their 14-year disease history (2002–2015). Importantly, the data for 2912 patients with pancreatic cancer were extracted from the NHISS DB using the corresponding KSCDCD cancer codes. Factors such as the extent of disabilities, diseases, death, and the current state of medical treatment were also considered.

### 3.2. Comparison between Cancer and Non-Cancer Groups in Both Sexes

A comparison of pancreatic-cancer-specific symptoms by sex is shown in Table 2. The sex-specific parameters of BMI, SBP, DBP, fasting blood glucose concentration, and total cholesterol concentration were significantly lower in women with pancreatic cancer than in women in the non-cancer group (*p* < 0.05). In men, the values of height, weight, SBP, DBP, hemoglobin, fasting blood glucose, serum glutamic oxaloacetic transaminase and aspartate aminotransferase (SGOT), and gamma glutamyl transpeptidase (GTP) concentrations were significantly lower, and total cholesterol concentration was higher in the cancer group than in the non-cancer group (*p* < 0.05; Table 2).

### 3.3. Sex Differences in Pancreatic Cancer Incidence

Sex-specific differences were also observed in this study. Smoking history, height, weight, SBP, hemoglobin, fasting blood glucose, total cholesterol, serum glutamic oxaloacetic transaminase and aspartate aminotransferase (SGOT), gamma glutamyl transpeptidase concentrations, and cancer history were found to be associated with pancreatic cancer in men (*p* < 0.05). In women, smoking history, height, weight, BMI, SBP, urine protein, hemoglobin, fasting blood glucose, total cholesterol, serum glutamic pyruvic transaminase and alanine aminotransaminase (SGPT), gamma glutamyl transpeptidase concentrations, and cancer history were associated with pancreatic carcinogenesis (*p* < 0.05; Table 3).

### 3.4. Sex Differences in Pancreatic Cancer Incidence by Exercise

We identified exercise-related parameters, including the type, intensity, duration, and frequency of exercise, which affected the risk of pancreatic cancer in a sex-dependent manner (Table 4).

Men had an approximately 77% decreased risk of pancreatic cancer (OR, 0.227; 95% CI, 0.089–0.582) when performing 30 min of daily exercise (EXERCI) compared to a 93.9% increased risk for women who exercised once or twice a week (OR 1.939; 95% CI, 1.231–3.055; *p* < 0.05).

Men showed an approximately 50% decrease in pancreatic cancer risk when performing 20 min of high-intensity exercise every day and more than a 100% increase in pancreatic cancer risk when undertaking 20 min of high-intensity exercise only once a week. However, women had at least a 100% increase in pancreatic cancer risk after performing 20 min of high-intensity exercise one to five times per week (*p* < 0.05).

Men undertaking 30 min of moderate-intensity exercise every day (MPA) showed a 46% decrease in pancreatic cancer risk, but those undertaking 30 min of moderate-intensity exercise once a week had an increased risk of at least 50%. Women who undertook moderate-intensity exercise one to five times per week showed an increased risk of at least 63%.

Men were more likely to develop pancreatic cancer if they undertook low-intensity walking (LPA) once a week (OR, 1.827; 95% CI, 1.179–2.830), and women were more likely to develop pancreatic cancer if they walked 1–5 times a week (*p* < 0.05).

### 3.5. Sex-Specific Obesity-Related Output from ROC Curve Analysis and the Summary of the Results

The cut-off values for the significant parameters identified in the logistic regression analysis were determined using a ROC curve analysis (Table 5). ROC curve analysis of pancreatic cancer provided the optimal threshold value in sex different way (Figure 2). The cut-off value for the total cholesterol concentration in men with pancreatic cancer was 188.50 mg/dL, which showed sensitivity of 53.5% and specificity of 54.6% (*p* < 0.05). The cut-off values for the significant variables identified in women with pancreatic cancer, based on ROC curve analysis, were a height of 165.50 cm, a weight of 58.50 kg, and a gamma glutamyl transpeptidase concentration of 20.50 U/L. The sensitivity and specificity ranges for these cut-off values were 57.3–76.3 and 57.4–75.8, respectively.

After analyzing the data for 1,108,369 individuals registered in the NHISS DB, the results obtained were summarized.

Of the 1,108,369 patients registered in the NHISS DB, 2912 patients with pancreatic cancer (903 men and 2009 women) were included in this study.BMI, SBP, DBP, and fasting blood glucose and total cholesterol levels were significantly lower in women with pancreatic cancer than those in the non-cancer group (*p* < 0.05). Meanwhile, the height, weight, hemoglobin, serum glutamic pyruvic transaminase and alanine aminotransaminase (SGPT), and gamma glutamyl transpeptidase (GTP) concentrations were lower in women with pancreatic cancer (*p* < 0.001). In men, the height, weight, SBP, DBP, hemoglobin, fasting blood glucose, total cholesterol, serum glutamic oxaloacetic transaminase and aspartate aminotransferase (SGOT), and gamma glutamyl transpeptidase concentrations were significantly different between the pancreatic cancer and non-cancer groups (*p* < 0.05).In the logistic regression analysis, a smoking history of <29 years had a higher OR (1.975–5.332) than other parameters in men, and women also had a higher pancreatic cancer incidence if they had a smoking history (OR, 8.936–18.330).Of all the types of exercise assessed, daily exercise was beneficial to lowering the risk of pancreatic cancer in men.The ROC curve analysis revealed that the total cholesterol concentration was the only significant factor associated with pancreatic cancer in men (*p* < 0.05). The ideal cut-off value for total cholesterol concentration was 188.50 mg/dL, which showed a sensitivity and specificity of 53.5% and 54.6%, respectively. However, height, weight, and gamma glutamyl transpeptidase concentration were the factors associated with pancreatic cancer in women, with ideal cut-off values of 165.50 cm, 58.50 kg, and 20.50 U/L, respectively. These cut-off values showed a sensitivity range of 57.3–76.3% and a specificity range of 57.4–75.8%.

## 4. Discussion

### 4.1. Related Results

Smoking accounts for 21% of all cancer fatalities, followed by overweight, obesity, and physical inactivity, which account for 2% [15]. Hence, in order to decrease pancreatic cancer risk, the following golden rule should be followed without exceptions: avoid smoking, prevent obesity, eat a balanced diet, and perform regular physical activity.

When comparing the pancreatic cancer and non-cancer groups using Student’s t-test, significant differences were observed in both sexes. The height, weight, blood pressure, hemoglobin, fasting blood glucose, total cholesterol, and gamma glutamyl transpeptidase concentrations were all significantly different in the cancer group (*p* < 0.01). In contrast, obesity-related factors, such as total cholesterol concentration, were significantly different in men, although pancreatic cancer is more prevalent in women than in men [16]. Renal failure might induce the hepatic microsomal enzyme to detoxify the abnormal circulating toxins, which increases the sensitivity of serum glutamic pyruvic transaminase and alanine aminotransaminase in the hepatic damage [17]. As such, the same mechanism can be suggested in the pancreas. Many serum markers, such as serum glutamic oxaloacetic transaminase and aspartate aminotransferase, gamma glutamyl transpeptidase, and hemoglobin, also showed different expressions between men and women in this study. Different efficacies of the immunotherapeutic approach between men and women had been evidenced [18,19,20]. The hormone differences or X and Y chromosome related immune genes affect different immune responses [21].

Moreover, logistic regression and ROC curve analyses of fasting blood glucose and total cholesterol concentrations (cut-off value, 188.50 mg/dL; sensitivity, 53.5%; specificity, 74.3%) in men with pancreatic cancer showed consistent results compared with those for women (*p* < 0.05). The cancer patients included in this study may have lipid-lowering medications, including hyperlipidemia and hypothyroidism, which could affect pre-operative total cholesterol concentrations. However, given that the average total cholesterol in teenagers is 156.6 mg/dL [22] and the average total cholesterol in Koreans (30~over 60 yrs) is 184.84 mg/dL [23], the results of cancer patients from this study (188.50 mg/dL) are higher than this, suggesting that the results are reliable. The findings in this study are consistent with those of previous studies on the association between diabetes and pancreatic cancer [24].

Our results indicate that obesity-related factors (fasting blood glucose and total cholesterol concentrations) may have a significant effect in patients with pancreatic cancer of both sexes. This suggests that various obesity-related etiological factors affect the risk of pancreatic cancer, and based on the results of this study, these may be attributed to endocrine secretion, diverse ethnicities, and lifestyle [25,26].

### 4.2. Association between Obesity and Pancreatic Cancer

According to Table 2, the mean total cholesterol concentration for men in the pancreatic cancer group was 196.40 mg/dL. The ROC curve analysis showed that the optimal threshold value for avoiding pancreatic cancer was 188.50 mg/dL (Table 3). A similar optimal threshold value for the total cholesterol concentration was not found in Korean men based on the NHISS DB.

Our results are consistent with those of previous studies, which have reported obesity as a marker of pancreatic cancer. For example, BMI is considered to be an influencing factor, which increases the incidence of leukemia and pancreatic, uterine, and colon cancers [27]. Adipokine-related signals from the intrapancreatic adipocyte exacerbate pancreatitis, which is well established as a risk factor for subsequent development of pancreatic cancer [28].

Our results further emphasize that obesity-related parameters in pancreatic cancer are sex-dependent (i.e., women had higher AUCs in the ROC analysis of height, weight, and gamma glutamyl transpeptidase concentration compared to men). Based on the results of this study, specific preventive interventions, such as tailored exercise programs, can be developed.

### 4.3. Effects of Physical Activity on Pancreatic Cancer via Obesity Modulation

Diverse methods, such as medication and surgery, have been used to treat pancreatic cancer. Several previous studies have reported that increased levels of adipose tissue activity cause overexpression of pro-inflammatory cytokines, hyperactivation of the insulin-like growth factor pathway, increased adipocyte-derived adipokine levels, hypercholesterolemia, and excessive oxidative stress, which contribute to carcinogenesis by regulating tumor behavior, inflammation, and the post-tumoral microenvironment [29,30].

A preclinical study on diverse aspects, such as quality of life, survival rate, behavioral analysis, and protein levels, in animal cancer models created using colon carcinoma cell lines unexpectedly found that intense oxidative exercise (rather than moderate-intensity exercise) gave the same results [31,32]. Our human-based results in men (daily intensive exercise decreased the risk of pancreatic cancer, with an OR of 0.453) are also consistent with those in another Jee’s study (under review process). Cytotoxic T cells and natural killer (NK) cells are important for defense against tumor cells, and these cells seem to be activated during exercise [33,34]. In particular, CD8+ T cells and NK cells, rather than CD4+ T cells, are significantly effective during acute exercise [35], suggesting that different exercise intensities specifically affect the immune response differently [36].

The current study showed that exercise had a significant effect on pancreatic cancer risk in women, and our logistic regression analysis showed that women had a higher incidence of pancreatic cancer than men for all exercise types, durations, and volumes (Table 4). These are similar to the results of previous studies [37] showing a sex-dependent effect on pancreatic cancer, suggesting that exercise is associated with a reduced incidence of pancreatic cancer in men but not in women. This report suggests that the counteracting effect of lower estrogen levels and the non-estrogen-mediated anti-cancer properties of physical activity may explain the lack of an association found in women, as exercise reduces estrogen levels, and an increase in estrogen levels is associated with the incidence of pancreatic cancer. However, further investigation is required to determine whether appropriate types of exercise can minimize the risk of pancreatic cancer.

In men, an exercise program, which maintains the total cholesterol concentration < 188.50 mg/dL, is recommended to reduce pancreatic cancer risk (Table 5). The ROC analysis results differed between women and men. The cut-off values for women were 165.50 cm (AUC, 0.837; sensitivity, 76.3%; specificity, 75.8%) for height, 58.50 kg (AUC, 0.693; sensitivity, 63.1%; specificity, 64.5%) for weight, and 20.50 U/L (AUC, 0.602; sensitivity, 57.3%; specificity, 57.4%) for gamma glutamyl transpeptidase concentration. These data can be utilized to control pancreatic cancer in women based on factors, which differ from those that are important in men.

The limitations of using this NHISS DB are as follows. Patients are registered in the NHISS DB based on their diagnosis or questionnaire completion. At this time, the patient’s situations (e.g., lifestyle, medical history, etc.) are reflected in this DB. Medical history records other than the 2002–2015 DB are not available if the patient does not answer or does not have a specific diagnosis in this study (i.e., the analysis was conducted to find this out). We therefore operatively determined that the analysis would be performed on the 1 million patients from 2002 to 2015 for the parameters described in Figure 1 (e.g., for pancreatic cancer, we would look at patients with 22 codes from Figure 1 versus acute nasopharyngitis with J00 as a control).

Although not included in the above operational definition, there is no doubt that the following factors are associated with the development of pancreatic cancer: surgical history (confirmed cancer surgery), chronic pancreatitis, pancreatic disease, and alcohol consumption. In the case of alcohol, this is true for high alcohol consumption (three or more drinks per day) but not for low to moderate alcohol consumption [38]. This raises the possibility that men are more likely to develop pancreatic cancer than women [2]. The duration of the exercise and its span are also important factors for the effect on pancreatic cancer. However, the NHISS DB does not include those related parameters, which may thus be a limitation of the NHISS DB analysis.

Of note, the symptoms of certain cancer types can be managed based on the intensity and frequency of exercise, which are necessary to ameliorate each cancer symptom. Therefore, interventions addressing the significant parameters found in this study (e.g., sex-specific differences and total cholesterol concentration), such as exercise programs, should be developed to reduce the incidence of pancreatic cancer. The findings of this study suggest the promotion of an adjusted exercise-oriented lifestyle to improve patient prognosis and manage pancreatic cancer symptoms.

## 5. Conclusions

This study produced the following novel findings in patients with pancreatic cancer.

BMI, SBP, DBP, fasting blood glucose, total cholesterol, serum glutamic oxaloacetic transaminase and aspartate aminotransferase (SGOT) concentrations, and age were lower in women with pancreatic cancer than in women in the corresponding non-cancer group (*p* < 0.05). In men, the height, weight, SBP, DBP, hemoglobin, fasting blood glucose, total cholesterol, serum glutamic oxaloacetic transaminase and aspartate aminotransferase (SGOT), and gamma glutamyl transpeptidase concentrations were significantly different between the pancreatic cancer and non-cancer groups (*p* < 0.05).A higher number of years of smoking and obesity-related parameters, such as total cholesterol concentration, are applicable only in men, and a higher number of years of smoking, height, weight, hemoglobin, serum glutamine pyruvic transaminase and alanine aminotransaminase (SGPT), and gamma glutamyl transpeptidase concentrations in women were associated with a higher incidence risk of pancreatic cancer (*p* < 0.05).High- or moderate-intensity daily exercise was associated with a lower risk of pancreatic cancer (*p* < 0.05).The ROC curve analysis revealed that total cholesterol concentration (*p* < 0.05) was the only significant factor associated with pancreatic cancer in men. The optimal cholesterol concentration cut-off value of 188.50 mg/dL showed a sensitivity and specificity of 53.5% and 54.6%, respectively. For women, the height, weight, and gamma glutamyl transpeptidase concentration had optimal cut-off values of 165.0 cm, 58.50 kg, and 20.50 U/L, respectively, for pancreatic cancer risk.

Based on our analysis of large-scale data from the NHISS DB, sex-specific differences were identified in the parameters affecting the risk of pancreatic cancer. Thus, identifying significant parameters can help develop individualized evidence-based prognostic and preventive programs (such as tailored exercise-based programs) for the treatment of pancreatic cancer.

## Figures and Tables

**Figure 1 cancers-15-05444-f001:**
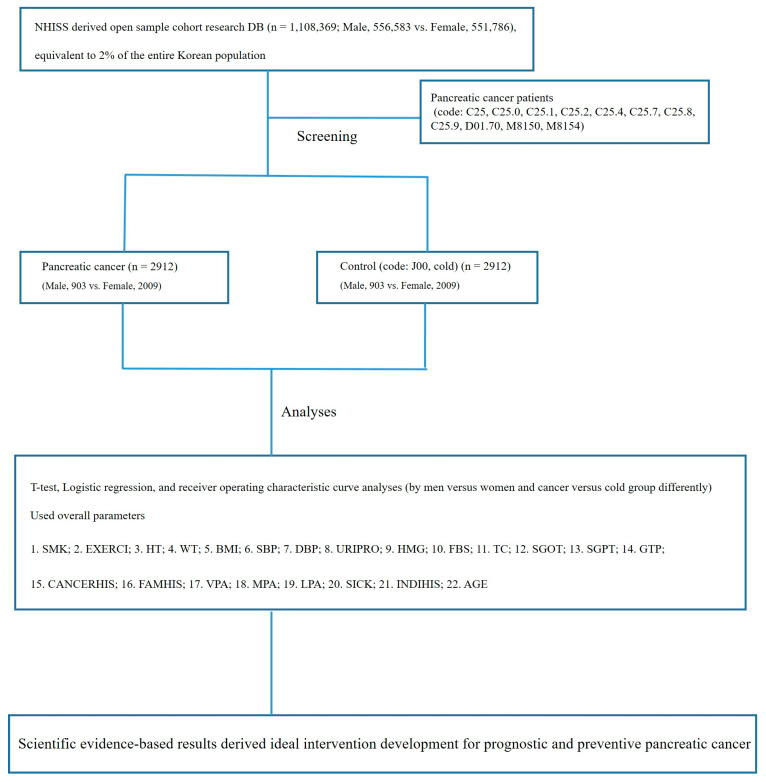
Data from 2912 patients with pancreatic cancer extracted from 1,108,369 patients registered in the NHISS DB and classified using KSCDCD codes (C25, C25.1, C25.2, C25.4, C25.7, C25.8, C25.9, D01.70, M8150, and M8154). Student’s *t*-tests, logistic regression, and receiver operating characteristic curve analyses were used to analyze sex-specific differences between cancer and non-cancer groups. NHISS DB, National Health Insurance Sharing Service Database; KSCDCD, Korean Standard Classification of Diseases and Causes of Death. SMK, smoking period in the past (1, 5–9 years; 2, 10–19 years; 3, 20–29 years; and 4, over 30 years); EXERCI, number of days of exercise (1, 1–2 times a week; 2, 3–4 times a week; 3, 5–6 times a week; and 4, almost every day); HT, height (cm); WT, weight (kg); BMI, body mass index (kg/m^2^); SBP, systolic blood pressure (mmHg); DBP, diastolic blood pressure (mmHg); URIPRO, protein in urine (1, weakly positive; 2, positive [+1]; 3, positive [+2]; 4, positive [+3]; and 5, positive [+4]); HMG level, hemoglobin level (g/dL); FBS, fasting blood sugar concentration (mg/dL); TC, total cholesterol concentration (mg/dL); SGOT, serum glutamic oxaloacetic transaminase and aspartate aminotransferase concentration (U/L); SGPT, serum glutamic pyruvic transaminase and alanine aminotransaminase concentrations (U/L); GTP, gamma glutamyl transpeptidase concentration (U/L); INDIHIS, individual disease history (1, tuberculosis; 2, hepatitis; 3, hepatism; 4, high blood pressure; 5, cardiopathy; 6, cerebral apoplexy; 7, diabetes; 8, cancer; and 9, others); FAMHIS, familial disease history (1, hepatism; 2, high blood pressure; 3, cerebral apoplexy; 4, cardiopathy; 5, diabetes; 6, cancer or others); VPA, number of days undertaking 20 min of intensive exercise in a week (1, 1 d; 2, 2 d; 3, 3 d; 4, 4 d; 5, 5 d; 6, 6 d; and 7, 7 d); MPA, number of days undertaking 30 min of moderate exercise in a week (1, 1 d; 2, 2 d; 3, 3 d; 4, 4 d; 5, 5 d; 6, 6 d; and 7, 7 d); LPA, number of days undertaking ≥ 30 min of walking in a week (1, 1 d; 2, 2 d; 3, 3 d; 4, 4 d; 5, 5 d; 6, 6 d; and 7, 7 d); SICK, sickness symptom; CANCERHIS, individual disease history (1, tuberculosis; 2, hepatitis; 3, hepatism; 4, high blood pressure; 5, cardiopathy; 6, cerebral apoplexy; 7, diabetes; 8, cancer; and 9, others); and AGE, age (years).

**Figure 2 cancers-15-05444-f002:**
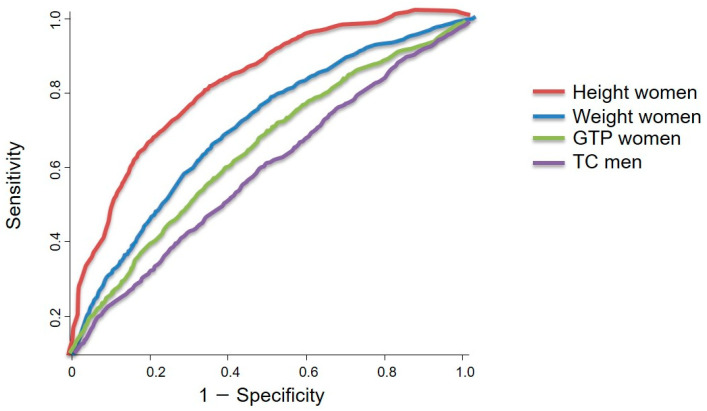
Graph of the ROC curve utilized in pancreatic cancer analysis. The AUC values for women (height, weight, and GTP concentration) and men (TC concentration) were 0.837, 0.693, 0.602, and 0.575, respectively, with cut-off values of 165.50 cm, 58.50 kg, 20.50 U/L, and 188.50 mg/dL, respectively. ROC, receiver operating characteristic; AUC, area under the curve; GTP, gamma glutamyl transpeptidase; TC, total cholesterol.

**Table 1 cancers-15-05444-t001:** Demographic characteristics of patients registered with the NHISS DB.

Parameters	Values
Whole registered population	1,108,369 (Men, 556,583; Women, 551,786)
Age (years)	0~85
Total follow-up years	2002–2015 (14 years)
Study contents	Disabilities, diseases, death, current state of medical treatment, etc.

NHISS DB, National Health Insurance Sharing Service Database.

**Table 2 cancers-15-05444-t002:** Sex differences in pancreatic cancer versus non-cancer groups.

	Men		Women	
	Cancer	Non-Cancer	Cancer	Non-Cancer
HT	163.60 ± 8.78 ***	165.60 ± 6.34	163.30 ± 9.06 ***	152.20 ± 6.42
WT	63.90 ± 11.79 *	65.14 ± 10.14	63.60 ± 12.08 ***	55.74 ± 9.06
BMI	23.78 ± 3.41	23.68 ± 2.96	23.75 ± 3.43 *	24.04 ± 3.43
SBP	122.60 ± 15.57 ***	127.50 ± 15.40	122.10 ± 14.42 ***	127.50 ± 16.12
DBP	76.40 ± 9.95 *	77.39 ± 10.40	76.12 ± 10.22 *	76.82 ± 9.68
HMG	14.07 ± 1.55 **	14.30 ± 15.60	13.99 ± 1.62 ***	12.76 ± 1.34
FBS	98.29 ± 23.56 ***	107.00 ± 29.21	97.68 ± 30.12 ***	103.20 ± 29.35
TC	196.40 ± 39.40 ***	185.20 ± 38.58	194.60 ± 6.52 ***	201.00 ± 41.04
SGOT	25.26 ± 13.64 ***	29.01 ± 19.75	25.39 ± 15.69	25.43 ± 12.65
SGPT	25.22 ± 24.36	26.78 ± 23.65	25.04 ± 21.07 ***	21.30 ± 13.98
GTP	35.10 ± 39.80 ***	47.65 ± 76.70	37.38 ± 56.61 ***	24.86 ± 27.46
AGE	56.69 ± 11.71	56.68 ± 11.70	57.95 ± 11.34	57.96 ± 11.34

Values are mean ± standard deviation. * *p* < 0.05, ** *p* < 0.01, *** *p* < 0.001: statistical significance between (pancreatic) cancer and non-cancer groups. HT, height (cm); WT, weight (kg); BMI, body mass index (kg/m^2^); SBP, systolic blood pressure (mmHg); DBP, diastolic blood pressure (mmHg); HMG, hemoglobin concentration (g/dL); FBS, fasting blood sugar concentration (mg/dL); TC, total cholesterol concentration (mg/dL); SGOT, serum glutamic oxaloacetic transaminase and aspartate aminotransferase concentrations (U/L); SGPT, serum glutamic pyruvic transaminase and alanine aminotransaminase concentrations (U/L); GTP, gamma glutamyl transpeptidase concentration (U/L); AGE, age (years).

**Table 3 cancers-15-05444-t003:** Logistic regression analysis of the parameters associated with pancreatic carcinogenesis.

Parameters	Men		Women	
	O.R.	95% CI	O.R.	95% CI
SMK (1)	5.332	1.821~15.612	15.954	6.928~36.742
SMK (2)	4.120	2.434~6.974	18.330	9.912~33.894
SMK (3)	1.975	1.304~2.993	11.580	6.641~20.191
SMK (4)	0.331	0.234~0.470	8.936	5.270~15.152
HT	0.967	0.955~0.979	1.197	1.183~1.212
WT	0.990	0.981~0.999	1.075	1.067~1.083
BMI	1.010	0.981~1.041	0.976	0.957~0.994
SBP	0.980	0.974~0.986	0.978	0.974~0.983
DBP	0.990	0.981~1.000	0.993	0.987~1.000
URIPRO (1)	1.122	0.595~2.117	0.769	0.494~1.198
URIPRO (2)	0.705	0.366~1.356	0.884	0.545~1.435
URIPRO (3)	0.948	0.342~2.627	0.176	0.067~0.463
URIPRO (4)	0.415	0.076~2.271	0.270	0.028~2.594
HMG	0.908	0.853~0.967	1.773	1.682~1.869
FBS	0.986	0.982~0.990	0.992	0.990~0.995
TC	1.007	1.005~1.010	0.996	0.994~0.997
SGOT	0.983	0.976~0.991	1.000	0.995~1.004
SGPT	0.997	0.993~1.001	1.014	1.010~1.019
GTP	0.996	0.993~0.998	1.013	1.010~1.016
CANCERHIS	0.374	0.257~0.543	0.345	0.273~0.434
FAMHIS	1.066	0.794~1.431	0.957	0.772~1.187

Variables with significant *p*-values were selected using logistic regression analysis. OR, odds ratio; CI, confidence interval; SMK, smoking period in the past (1, 5–9 years; 2, 10–19 years; 3, 20–29 years; and 4, >30 years); HT, height (cm); WT, weight (kg); BMI, body mass index (kg/m^2^); SBP, systolic blood pressure (mmHg); DBP, diastolic blood pressure (mmHg); URIPRO, protein in urine (1, weak positive; 2, positive [+1]; 3, positive [+2]; and 4, positive [+3]); HMG, hemoglobin concentration (g/dL); FBS, fasting blood sugar concentration (mg/dL); TC, total cholesterol concentration (mg/dL); SGOT, serum glutamic oxaloacetic transaminase and aspartate aminotransferase concentrations (U/L); SGPT, serum glutamic pyruvic transaminase and alanine aminotransaminase concentrations (U/L); GTP, gamma glutamyl transpeptidase concentration (U/L); CANCERHIS, individual disease history (1, tuberculosis; 2, hepatitis; 3, hepatism; 4, high blood pressure; 5, cardiopathy; 6, cerebral apoplexy; 7, diabetes; 8, cancer; and 9, others); FAMHIS, familial disease history, (1, hepatism; 2, high blood pressure; 3, cerebral apoplexy; 4, cardiopathy; 5, diabetes; 6, cancer or others). The references for SMK and URIPRO were less than 5 years old and negative (−), respectively.

**Table 4 cancers-15-05444-t004:** Logistic regression analysis of the exercise-related parameters on pancreatic carcinogenesis.

Parameters	Men		Women	
	O.R.	95% CI	O.R.	95% CI
EXERCI (1)	1.294	0.661~2.534	1.939	1.231~3.055
EXERCI (2)	1.264	0.482~3.313	1.826	0.863~3.862
EXERCI (3)	2.123	0.253~17.816	4.014	0.512~31.439
EXERCI (4)	0.227	0.089~0.582	1.014	0.459~2.243
VPA (1)	2.002	1.460~2.745	2.795	2.150~3.634
VPA (2)	1.151	0.783~1.692	2.969	2.197~4.011
VPA (3)	1.282	0.825~1.992	2.325	1.704~3.173
VPA (4)	0.865	0.455~1.644	1.772	1.107~2.837
VPA (5)	1.638	0.908~2.958	2.407	1.487~3.896
VPA (6)	0.540	0.203~1.436	1.760	1.000~3.098
VPA (7)	0.453	0.230~0.896	1.173	0.704~1.957
MPA (1)	1.568	1.121~2.194	2.728	2.112~3.522
MPA (2)	1.246	0.871~1.784	2.520	1.939~3.275
MPA (3)	0.915	0.603~1.387	1.762	1.332~2.332
MPA (4)	0.937	0.552~1.589	1.628	1.096~2.417
MPA (5)	1.071	0.632~1.814	1.705	1.118~2.599
MPA (6)	0.494	0.245~0.995	1.465	0.876~2.452
MPA (7)	0.535	0.314~0.912	0.804	0.521~1.240
LPA (1)	1.827	1.179~2.830	2.179	1.651~2.876
LPA (2)	1.421	0.980~2.060	2.179	1.688~2.812
LPA (3)	1.375	0.963~1.963	1.469	1.152~1.873
LPA (4)	1.486	0.927~2.385	1.598	1.184~2.157
LPA (5)	1.001	0.682~1.470	1.704	1.302~2.231
LPA (6)	1.011	0.628~1.626	1.362	0.994~1.866
LPA (7)	0.767	0.545~1.081	1.130	0.886~1.441

Variables with significant *p*-values were selected via logistic regression analysis. O.R., odds ratio; C.I., confidence interval; EXERCI, number of dates with exercise (1, 1-2 times in a week; 2, 3–4 times in a week; 3, 5–6 times in a week; and 4, almost everyday); VPA, number of dates with 20 min of intensive exercise in a week (1, 1 d; 2, 2 d; 3, 3 d; 4, 4 d; 5, 5 d; 6, 6 d; and 7, 7 d); MPA, number of dates with 30 min of moderate exercise in a week (1, 1 d; 2, 2 d; 3, 3 d; 4, 4 d; 5, 5 d; 6, 6 d; and 7, 7 d); LPA, number dates with ≥ 30 min of walking exercise in a week (1, 1 d; 2, 2 d; 3, 3 d; 4, 4 d; 5, 5 d; 6, 6 d; and 7, 7 d). The reference of EXERCI, VPA, MPA, and LPA is none (no exercise).

**Table 5 cancers-15-05444-t005:** ROC curve analyses for males and females with pancreatic cancer.

	Male				Female			
	AUC	Cut-Off Values	Sensitivity (%)	Specificity (%)	AUC	Cut-Off Values	Sensitivity (%)	Specificity (%)
HT	0.429	164.50	46.3	43.8	0.837	165.50	76.3	75.8
WT	0.454	63.50	47.7	43.3	0.693	58.50	63.1	64.5
TC	0.575	188.50	53.5	54.6	0.456	195.50	46.4	47.5
GTP	0.398	25.50	43.0	41.9	0.602	20.50	57.3	57.4

Parameters in males and females with significant *p*-values were obtained using logistic regression analysis, and the optimal cut-off points were determined via receiver operating characteristic curve analysis. AUC, area under the curve; HT, height (cm); WT, weight (kg); TC, total cholesterol concentration (mg/dL); GTP, gamma glutamyl transpeptidase concentration (U/L).

## Data Availability

Publicly available datasets were analyzed in this study. These data can be found at http://nhiss.nhis.or.kr.

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
