# Peer review of "Determining Optimal Cut-Off Value of Pancreatic-Cancer-Induced Total Cholesterol and Obesity-Related Factors for Developing Exercise Intervention: Big Data Analysis of National Health Insurance Sharing Service Data"

_cancers, 2023, doi:10.3390/cancers15225444_

Round 1

Reviewer 1 Report

Comments and Suggestions for Authors

The study utilised a large Korean database (the National Health Insurance Sharing Service (NHISS DB)).  The study explored blood-based markers plus some risk factors such as smoking BMI etc.  I have some comments over the methodology.

1. line 85-87, the authors used the word randomly selected but in the diagram, it showed full figure.  The authors need to make it clear how the random selection was carried out and why.  Numbers also need to be clear after random selection.

2.  It is not clear to me about time line of data for example at the beginning when they registered to the database or over time.  Exposure time prior to diagnosis is a key component to case-control analysis.  The authors should explain very clearly what time point of the data were used.  This will lead to point 3.

3. Some biomarkers changed overtime therefore , this point needs to be discussed thoroughly.  Pancreatitis is also a key risk factor given that you can extract data from ICD codes, this should also be explored.  Alcohol intake also a potential factor, if the authors do not have the data, this should be mentioned in the discussion.

4.  In the conclusion section, there are many serum markers show significantly difference in men and women.  These serums and their role have not been discussed in the discussion part.  

5. Exercise showed risk reduction however time captured exercise was not mentioned (average of a long period? or a one snapshot of the data).

Author Response

  1. line 85-87, the authors used the word randomly selected but in the diagram, it showed full figure.  The authors need to make it clear how the random selection was carried out and why.  Numbers also need to be clear after random selection.

[Response]  

Thank you for your valuable advice. Here is my answer to your query.

If an applicant sign up for the Cohort (not by an applicant’s choice), NHISS will randomly access the approximately 1 million cohort population.

The cohort data is a nationally representative sample of the population with a baseline of 2002, providing prospective (2002-2015) medical history information for the entire sample.

We also added the following sentences in 2. Materials and Methods / 2.1 Data source and subject population

The DB is a sample cohort data for research purposes collected and accumulated from existing nationwide surveys and hospital prescription results. For this study, we used publicly available data that has been fully surveyed and finalized into a dataset by applying inclusion/exclusion criteria to the 1 million cohort data. The National Health Insurance Service DB is not customized data centered on specific diseases, but a DB sampled from the population that received health checkups regardless of disease, i.e., it is a sample data of the 1 million cohort, so it is a sample data of the entire population. The sample survey DB contains 1,108,369 patient records, which is about 2% of the total population of South Korea. These 1,108,369 randomly selected patients are designed to be statistically representative of the entire population of South Korea (approximately 50 million people) and can be filtered by gender, age, medical insurance, region, and other strata. The number of people in the sample cohort data may vary slightly, as there is a 2% oversampling to reflect additional deaths and newborns each time.

  1. It is not clear to me about time line of data for example at the beginning when they registered to the database or over time.  Exposure time prior to diagnosis is a key component to case-control analysis.  The authors should explain very clearly what time point of the data were used.  This will lead to point 3.

 [Response]  

Thank you for your precious advice. I added the below phrases in the end of Discussion section (4.3 Effects of physical activity on pancreatic cancer via obesity modulation).

The limitations of using this NHISS DB are as follows.

Patients are registered in the NHISS DB based on their diagnosis or questionnaire completion. At this time, the patient's situations (e.g., lifestyle, medical history, etc.) are reflected in this DB, Medical history records other than the 2002-2015 DB are not available if the patient does not answer or does not have a specific diagnosis in this study. (i.e., the analysis was conducted to find out this).

We therefore operatively defined that the analysis would be performed on the 1 million patients from 2002 to 2015 for described parameters in Figure 1 (e.g., for pancreatic cancer, we would look at patients with 22 codes from Figure 1 vs. acute nasopharyngitis with J00 as a control).

  1. Some biomarkers changed overtime therefore , this point needs to be discussed thoroughly.  Pancreatitis is also a key risk factor given that you can extract data from ICD codes, this should also be explored.  Alcohol intake also a potential factor, if the authors do not have the data, this should be mentioned in the discussion.

 [Response]  

Thank you for your precious comments.

As mentioned above query 2, we used an operational definition, so other analyses are not possible at this time.

The reason is that it takes about 6 months to 1 year to access the NHISS DB for the current analysis, and if we proceed, we will have to report it in a new paper.

We added the following phrases in the Discussion section (4.2. Association between obesity and pancreatic cancer).

Similar to the consequences of obesity-related factors, adipokine related signals from inctrapancreatic adipocyte exacerbate pancreatitis which is  well-established as a risk factor for subsequent development of pancreatic cancer.

  1. In the conclusion section, there are many serum markers show significantly difference in men and women.  These serums and their role have not been discussed in the discussion part.  

 [Response]  

Thank you for your valuable comments. We added the below phrases in the Discussion section (4.1 Related results)

Renal failure might induce hepatic microsomal enzyme to detoxify the abnormal circulating toxins, which increases the sensitivity of serum glutamic pyruvic transaminase and alanine aminotransaminase in the hepatic damage. As such, the same mechanism can be suggested in the pancreas. Many serum markers such as serum glutamic oxaloacetic transaminase and aspartate aminotransferase, gamma glutamyl transpeptidase, and hemoglobin also showed different expressions between men and women in his study. Different efficacy of immunotherapeutic approach between men and women had been tried. Hormone differences or X and Y chromosome related immune genes affect different immune response.

  1. Exercise showed risk reduction however time captured exercise was not mentioned (average of a long period? or a one snapshot of the data).

 [Response]  

Thank you for your comments. According to your precious comments, we added the following phrases in the end of Discussion section (4.3 Effect of physical activity on pancreatic cancer via obesity modulation).

The duration of exercise and its span are also important factors for the effect, however, the NHISS DB does not include those related parameters, this may be thus a limitation of the NHISS DB analysis.

Reviewer 2 Report

Comments and Suggestions for Authors

Jee et al. performed an interesting research on the risk of pancreatic cancer (PC). They not only identified the factor associating with PC occurrence in men and women, but also explored the effect of physical exercise pattern on the PC occurrence and determined optimal cut-off value of total cholesterol (TC) concentration in preventing PC occurrence. Currently obesity and diabetes has been acknowledged as the risk factor for PC and may benefit from exercise. As a large-sample research, this research confirmed the clinical value of exercise and proposing the optimal intensity of exercise in preventing PC occurrence, which is practical and of clinical significance. They also proposed the optimal cut-off value of TC concentration which can be applied as the target of daily exercise to avoid PC. Therefore this research is of certain novelty and clinical value. Here are my suggestions:   

1. In this research, authors not only determined the optimal cut-off value of TC concentration, but also researched the obesity factors associating with PC incidence and the value of exercise. Current title only included a part of this research. It will be better to rearrange the title and included exercise in title.  

2. As development of surgical technique and medication therapy, the 5-year survival rate of pancreatic cancer patients increased in recent years. The 5-year survival rate showed in line 54-55 is less accurate. It will be better to cite recent literature and update the survival rate.

3. From line 60-62, the author cited a research indicating that pancreatic cancer patients with higher BMI have higher risk of colon cancer. However, after going through the cited research, the didnt report such result. Please go through the origin article and confirm if the result is mistakenly written.  

4. In terms of the patients inclusion, pre-operative TC concentration can be affected by a series of disease including hyperlipemia, hypothyroidsm as well as the medication including lipid-lowering drugs. Whether patients with these diseases and medical treatment are included in this cohort? If included, it will be better to exclude these patients so as to make conclusion more convincing. If not, these should be mentioned in the patients inclusion and exclusion part.  

5. Please rearrange figure 1. Pancreatic cancer group and control group can be arranged parallelly instead of sequentially, besides the patients number and gender constitution should be included.

6. Surgical history including subtotal gastrectomy, chronic pancreatic diseases including chronic pancreatitis and alcohol consumption have been accept as the risk factors for PC occurrence. Therefore, history of gastric surgery, chronic pancreatitis and alcohol consumption can also be included in logistic regression analyzing risk factors of PC incidence.

7. It will be better to include p value in table 3.

8. Line 260-285 in discussion part should be moved to result part.

9. In line 379-380, it should be emphasized that conclusion 3 in only applicable in man.

10. In line 381-386, the optimal cut-off value of serum TC concentration and its specificity and sensitivity in men are different from that in the Result part, so as the cut-off value of height in women. Authors have to check it and confirmed if its wrongly written. 

Author Response

  1. In this research, authors not only determined the optimal cut-off value of TC concentration, but also researched the obesity factors associating with PC incidence and the value of exercise. Current title only included a part of this research. It will be better to rearrange the title and included exercise in title.

 [Response]  Thank you for your valuable comments.

We rearranged the title as follows,

Determining optimal cutoff value of pancreatic cancer-induced total cholesterol and obesity related factors for developing exercise interventions: Big-data analysis of National Health Insurance Sharing Service Data

  1. As development of surgical technique and medication therapy, the 5-year survival rate of pancreatic cancer patients increased in recent years. The 5-year survival rate showed in line 54-55 is less accurate. It will be better to cite recent literature and update the survival rate.

 [Response]  

Thank you for your precious comments. We replaced the following sentence in the beginning of the Introduction.

The 5-year survival rate for pancreatic cancer ranges from 2% to 9%, depending on the country, with regional and income differences, however, does not exceeds 10%.

  1. From line 60-62, the author cited a research indicating that pancreatic cancer patients with higher BMI have higher risk of colon cancer. However, after going through the cited research, the didn’t report such result. Please go through the origin article and confirm if the result is mistakenly written.

 [Response]  

Thank you for your precious advice. We corrected the following sentence.

A previous study reported that free of prior cancer at baseline with a body mass index (BMI)≥30 kg/m2  have about 70% higher risk of pancreatic cancer, compared to those with a BMI < 23 kg/m2 [11],

  1. In terms of the patients inclusion, pre-operative TC concentration can be affected by a series of disease including hyperlipemia, hypothyroidsm as well as the medication including lipid-lowering drugs. Whether patients with these diseases and medical treatment are included in this cohort? If included, it will be better to exclude these patients so as to make conclusion more convincing. If not, these should be mentioned in the patients inclusion and exclusion part.

 [Response]  

Thank you for your precious comments.

As a result, the average TC for Koreans aged 10 to 18 is 156.6 mg/dL.

And, as you can see in the below screen shot from Google search, if you search for "normal total cholesterol(=>total cholesterol 정상수치)" in Koreans, the average is about 180 mg/dL, as shown.

It can be said that the sample DB is higher than the above average for cancer patients (of course it is higher for cancer patients), and we can also think about the effect of error because the DB is a cohort DB corresponding to 2% of Korean population rather than the whole population.

We added the following phrases in the Beginning of Discussion section (4.1 Related results)

The cancer patients included in this study may have lipid-lowering medications, including hyperlipidemia and hypothyroidism, which could affect pre-operative total cholesterol concentrations. However, given that the average total cholesterol in teenagers is 156.6 mg/dL and the average total cholesterol in Koreans (30 ~ over 60 yrs) is 184.84 mg/dL, the results of cancer patients from this study (188.50 mg/dL) are higher than this, suggesting that the results are reliable.

  1. Please rearrange figure 1. Pancreatic cancer group and control group can be arranged parallelly instead of sequentially, besides the patients number and gender constitution should be included.

 [Response]  

Thank you for your precious advice. We changed the figure 1 as shown in the manuscript. Thank you very much.

  1. Surgical history including subtotal gastrectomy, chronic pancreatic diseases including chronic pancreatitis and alcohol consumption have been accept as the risk factors for PC occurrence. Therefore, history of gastric surgery, chronic pancreatitis and alcohol consumption can also be included in logistic regression analyzing risk factors of PC incidence.

 [Response]  

Thank you for your precious comments. We added the following phrases in the end of Discussion (4.3 Effects of physical activity on pancreatic cancer via obesity modulation)

Although not included in the above operational definition, there is no doubt that the following factors are associated with the development of pancreatic cancer: surgical history (confirmed cancer surgery), chronic pancreatitis, pancreatic disease, and alcohol consumption. In the case of alcohol, this is true for high alcohol consumption (3 or more drinks per day), but not for low to moderate alcohol consumption. This raises the possibility that men are more likely to develop pancreatic cancer than women.

  1. It will be better to include p value in table 3.

 [Response]  We appreciate very much for your precious advice.

According to your valuable comments, we would like to answer as follows that

the p-value is significant if the confidence interval does not include 1. Since this is a 1 million cohort DB, the number is large, so as number increases, the significance of p-vaue decreases and can approach zero. However, the confidence interval is not affected by the number, so we can say that it is more reliable. Therefore, the significance of the multivariate regression analysis is determined by looking at the confidence interval rather than the p-value, and the p-value is excluded for this reason.

  1. Line 260-285 in discussion part should be moved to result part.

 [Response]  

Thank you for your valuable advice. We moved the phrases in the end of Results section in the corrected subtitle (3.5. Sex-specific obesity-related output from ROC curve analysis and the summary of the results)

3.5. Sex-specific obesity-related output from ROC curve analysis and the summary of the results

  • •••••

After analyzing the data for 1,108,369 individuals registered in the NHISS DB, the obtained results were summarized.

  1. Of the 1,108,369 patients registered in the NHISS DB, 2,912 with pancreatic cancer (903 men and 2,009 women) were included in this study.
  2. BMI, SBP and DBP, and fasting blood glucose and total cholesterol levels were significantly lower in women with pancreatic cancer than those in the non-cancer group (p < 0.05). Meanwhile, height; weight; and hemoglobin, serum glutamic pyruvic transaminase and alanine aminotransaminase, and gamma glutamyl transpeptidase concentrations were lower in women with pancreatic cancer (p < 0.001). In men, height; weight; SBP and DBP; and hemoglobin, fasting blood glucose, total cholesterol, serum glutamic oxaloacetic transaminase and aspartate aminotransferase, and gamma-glutamyl transpeptidase concentrations were significantly different between the pancreatic cancer and non-cancer groups (p < 0.05).
  3. In logistic regression analysis, a smoking history < 29 years had a higher OR (1.975–5.332) than other parameters in men, and women also had a higher pancreatic cancer incidence if they had a smoking history (OR, 8.936–18.330).
  4. Of all types of exercise assessed, daily exercise was beneficial to lowering the risk of pancreatic cancer in men.
  5. ROC curve analysis revealed that the total cholesterol concentration was the only significant factor associated with pancreatic cancer in men (p < 0.05). The ideal cutoff value for total cholesterol concentration was 188.50 mg/dL, which showed a sensitivity and specificity of 53.5% and 54.6%, respectively. However, height, weight, and gamma glutamyl transpeptidase concentration were the factors associated with pancreatic cancer in women, with ideal cutoff values of 165.50 cm, 58.50 kg, and 20.50 U/L, respectively. These cutoff values showed a sensitivity range of 57.3–76.3% and a specificity range of 57.4–75.8%.

  1. In line 379-380, it should be emphasized that conclusion 3 in only applicable in man.

 [Response]  

Thank you for your valuable comments. We corrected as following sentence.

A higher number of years of smoking and obesity-related parameters, such as total cholesterol concentration are applicable only in men

  1. In line 381-386, the optimal cut-off value of serum TC concentration and its specificity and sensitivity in men are different from that in the Result part, so as the cut-off value of height in women. Authors have to check it and confirmed if it’s wrongly written.

 [Response]  

Thank you for the correction. We've made the changes you mentioned,

And we've highlighted all corrected in yellow throughout the manuscript. Thank you very much again.

Round 2

Reviewer 1 Report

Comments and Suggestions for Authors

Authors addressed my comments and i am satisfy with the answers.